# DOMAIN-INVARIANT REPRESENTATIONS: A LOOK ON COMPRESSION AND WEIGHTS

## ABSTRACT

Learning *Invariant Representations* to adapt deep classifiers of a source domain to a new target domain has recently attracted much attention. In this paper, we show that the search for invariance favors the compression of representations. We point out this may have a bad impact on adaptability of representations expressed as a minimal combined domain error. By considering the risk of compression, we show that weighting representations can align representation distributions without impacting their adaptability. This supports the claim that representation invariance is too strict a constraint. First, we introduce a new bound on the target risk that reveals a trade-off between compression and invariance of learned representations. More precisely, our results show that the adaptability of a representation can be better controlled when the compression risk is taken into account. In contrast, preserving adaptability may overestimate the risk of compression that makes the bound impracticable. We support these statements with a theoretical analysis illustrated on a standard domain adaptation benchmark. Second, we show that learning weighted representations plays a key role in relaxing the constraint of invariance and then preserving the risk of compression. Taking advantage of this trade-off may open up promising directions for the design of new adaptation methods.

## 1 INTRODUCTION

In a standard *Supervised Learning* setup, we consider that training data is representative of testing data. This ideal framework is often violated in practical applications since the data generative process can be altered during the application phase *e.g.* different pre-processing, sample rejection, imbalanced classes, condition of data collection, time evolving data... This so-called *Distributional Shift* situation (Quionero-Candela et al. (2009); Kull & Flach (2014)) is recognized as a major challenge in Machine Learning (Amodei et al. (2016)). *Domain Adaptation* (DA) (Pan & Yang (2009)) is a widely adopted strategy to mitigate its effects. It consists in leveraging unlabeled testing data (the *target domain*) in order to transfer knowledge learned on labeled training data (the *source domain*).

A first line of study, named *Importance Sampling* (IS), assumes some stationarity in distributions in order to approximate the risk in the target domain by weighting sample contribution (Shimodaira (2000); Huang et al. (2007)). To be applied, IS needs statistical support sharing across domains limiting its range of application when addressing high dimensional data. In that challenging context, Ganin & Lempitsky (2015) have shown that deep classifiers can be adapted to a new target domain by learning domain *Invariant Representations* (IR) and find theoretical support in the work of Ben-David et al. (2007; 2010). It assumes that a classifier trained on representations of labeled data sampled from both domains will achieve a negligible combined error (*adaptability*). However, enforcing distribution invariance in the representation space can badly impact its adaptability, for instance in the context of *Target Shift* (Zhao et al. (2019)) or when the support of the distributions do not overlap enough (Johansson et al. (2019)). In the most general case of DA, it is not clear how IR helps aligning domains and removing spurious correlations as pointed by Arjovsky et al. (2019).

In the present work, we show that the search of invariance favors learning representations with a significant loss of information from the original features. We refer to this phenomenon as the risk of *Compression*. In our understanding, this risk is not addressed in the original work from Ben-David et al. (2007; 2010). We demonstrate that taking into account this risk has two major advantages. First, it provides a better control on (non-trainable) terms involving the labels in the

target domain. Second, it introduces a new trainable term which embodies a risk of *Compression* and then may open-up the path to new learning methods. We provide a theoretical analysis illustrated with experiments on a standard Domain Adaptation benchmark. We underline that a better control on non-trainable terms may lead to an over-pessimistic risk of compression. This brings to light an inherent trade-off when learning representations for domain adaptation: *How much compression risk is acceptable in order to obtain invariance?* As a first try to answer to this question, we point that *Learning Weighted Representations* plays a key role in aligning representation distributions in order to obtain invariance. More precisely, we show that under a constraint of *strong source risk conservation*, weighting representations does not impact its adaptability.

## 2 BACKGROUND

### 2.1 A DECADE OF DOMAIN ADAPTATION

The classical setup for modelling distributional shift is to introduce two domains: the *source* domain *i.e.* dataset where the model is trained with supervision and the *target* domain *i.e.* where the model is tested or applied. Formally, for two random variables $(X, Y)$ of a given space $\mathcal{X} \times \mathcal{Y}$, we introduce two distributions on $\mathcal{X} \times \mathcal{Y}$: the source distribution $\mathbb{P}^s(X, Y)$ and the target distribution $\mathbb{P}^t(X, Y)$. We use the exponent notation $s$ and $t$ to differentiate source and target terms. We define the hypothesis space $\mathcal{H}$ as a subset of functions from $\mathcal{X}$ to $\mathcal{Y}$. The distributional shift situation is then characterized by the shift: $\mathbb{P}^s(X, Y) \neq \mathbb{P}^t(X, Y)$. Learning under *Distributional Shift* (Quionero-Candela et al. (2009)) consists in minimizing the risk on the target domain $\varepsilon^t(h) = \mathbb{E}^t[\ell(h(X), Y)]$ for a given loss $\ell : \mathcal{Y} \times \mathcal{Y} \to \mathbb{R}^+$ and an hypothesis function $h \in \mathcal{H}$. Assuming that $w(X, Y) = \mathbb{P}^t(X, Y)/\mathbb{P}^s(X, Y)$ is tractable, it is straightforward to show that:

$$\varepsilon^t(h) = \mathbb{E}^s[w(X, Y)\ell(h(X), Y)] \tag{1}$$

Roughly speaking, this equation shows that it is possible to minimize the target risk using data $(x_i^s, y_i^s)_{1 \leq i \leq n^s}$ sampled from the source domain (*i.e.* $(x_i, y_i) \sim \mathbb{P}^s(X, Y)$) while re-weighting their contribution in the source expectation with a factor $w(x_i, y_i)$ (*Importance Sampling*). The difficulty of *Learning under distributional shift* arises when $w(x_i, y_i)$ is not tractable. The first reason is that data sampled from $\mathbb{P}^t(X, Y)$ is often not easily available. In order to get a reasonable estimation of $w$, practitioners are used to reformulate the problem as *Unsupervised Domain Adaptation*.

**Definition 1** (Unsupervised Domain Adaptation). *Given a loss $\ell$, an hypothesis class $\mathcal{H}$, a source domain $(\mathcal{X} \times \mathcal{Y}, \mathbb{P}^s(X, Y))$ and a target domain $(\mathcal{X} \times \mathcal{Y}, \mathbb{P}^t(X, Y))$, Unsupervised Domain Adaptation consists in minimizing the target risk (i.e. finding $h^\star = \arg\min_{h \in \mathcal{H}} \mathbb{E}^t[\ell(h(X), Y)]$) with finite **labeled** sampling of $(x_i^s, y_i^s) \sim \mathbb{P}^s(X, Y)$ of the source domain and a finite **unlabeled** sampling $(x_j^t) \sim \mathbb{P}^t(X)$ of the target domain.*

If unlabeled samples in the target domain are available, a first idea is then to approximate $w(X, Y)$ by $w(X) = \mathbb{P}^t(X)/\mathbb{P}^s(X)$. This is equivalent to assuming that conditional distributions are conserved across domains (*i.e.* $\mathbb{P}^s(Y|X) = \mathbb{P}^t(Y|X)$), a situation known as *Covariate Shift*, that has already been the subject of an extensive literature (Sugiyama et al. (2008); Huang et al. (2007)). This formulation becomes ill-posed when feature domains $(\mathcal{X}, \mathbb{P}^s(X))$ and $(\mathcal{X}, \mathbb{P}^t(X))$ suffer from non-overlapping support: $w(X)$ is then undefined. To address this issue, *Invariant Representations* learning (Long et al. (2015)) performs adaptation by looking for a representation $Z = \varphi(X)$ whose distribution is conserved across domains $\mathbb{P}^s(Z) = \mathbb{P}^t(Z)$. A common strategy consists in extracting features $Z$ to learn labels $Y$ in the source domain while controlling representation invariance. That is formulated as follows:

**Learning Objective 1** (Learning Invariant Representations for Domain Adaptation).

$$g^\star, \varphi^\star = \arg \min_{(g, \varphi) \in \mathcal{G} \times \Phi} (1 - \alpha) \cdot \varepsilon^s(g \circ \varphi) + \alpha \cdot \mathbb{D}^{(s,t)}(\varphi) \tag{2}$$

*where $\mathcal{G}$ is the set of classifiers and $\Phi$ is the set of representations. The hypothesis space is then defined as $\mathcal{H} = \{g \circ \varphi : g \in \mathcal{G}, \varphi \in \Phi\}$. $\mathbb{D}^{(s,t)}(\varphi)$ is a distance between distributions $\mathbb{P}^s(Z)$ and $\mathbb{P}^t(Z)$ and $\alpha$ is a trade-off parameter.*

The choice of $\mathbb{D}^{(s,t)}(\varphi)$ has been the object of an extensive literature including $f-$ divergence measures such as domain adversarial learning (Ganin & Lempitsky (2015); Tzeng et al. (2014); Long

et al. (2018)), Integral Probability Measures such as *Maximum Mean Discrepancy* (Long et al. (2015; 2016)) or *Optimal Transport* (Shen et al. (2018)). See Kouw & Loog (2019) for an extensive review. Interestingly, learning invariant representations to perform adaptation finds theoretical support from the work of Ben-David et al. (2007; 2010) that we recall in the next section.

## 2.2 THEORETICAL GUARANTEES OF DOMAIN ADAPTATION

We consider the set of representations $\Phi$ and classifiers $\mathcal{G}$. The hypothesis space is denoted $\mathcal{H} = \{g \circ \varphi : g \in \mathcal{G}, \varphi \in \Phi\}$. We introduce the loss $\ell : \mathcal{Y} \times \mathcal{Y} \to \mathbb{R}^+$ which is assumed symmetric and verifies the triangular inequality. We note the risk $\varepsilon(h) = \mathbb{E}[\ell(Y, h(X))]$ for $h \in \mathcal{H}$ and $\varepsilon(h, h') = \mathbb{E}[\ell(h(X), h'(X))]$ for $(h, h') \in \mathcal{H}^2$. To emphasize the role of a representation $Z = \varphi(X)$, we introduce $\tilde{f}(Z) = \mathbb{E}[Y|Z]$ and $\tilde{\mathcal{H}} = \{g \circ \varphi : g \in \mathcal{G}\} \subset \mathcal{H}$. We underline that $\tilde{\cdot}$ quantities depend on $\varphi$ where, for the ease of reading, this dependence is omitted in notations when it is not ambiguous. Ben-David et al. (2007; 2010) have derived a theoretical bound of the additional risk when using a representation $Z = \varphi(X)$ rather than raw features $X$:

**Inequality 1** (Ben-David et al. (2007; 2010)). *For a given $\varphi \in \Phi$ and $g \in \mathcal{G}$, there is:*

$$\underbrace{\varepsilon^t(g \circ \varphi)}_{\text{Target risk}} \leq \underbrace{\varepsilon^s(g \circ \varphi)}_{\text{Source risk}} + \underbrace{\sup_{(h,h') \in \tilde{\mathcal{H}}} \left| \varepsilon^s(h, h') - \varepsilon^t(h, h') \right|}_{=\frac{1}{2} \cdot d(\tilde{\mathcal{H}} \Delta \tilde{\mathcal{H}}): \text{ Disagreement}} + \underbrace{\inf_{h \in \tilde{\mathcal{H}}} \varepsilon^t(h) + \varepsilon^s(h)}_{=\lambda(\tilde{\mathcal{H}}): \text{ Adaptability}} \tag{3}$$

The inequality 1 ensures that the target risk is bounded by the sum of the source risk, the disagreement risk between two classifiers from representations ($d(\tilde{\mathcal{H}} \Delta \tilde{\mathcal{H}})$ named $\tilde{\mathcal{H}} \Delta \tilde{\mathcal{H}}$ distance), and a third term ($\lambda(\tilde{\mathcal{H}})$) which quantifies the ability to perform well in both domains from representations. In the rest of the paper, we refer to the latter as the *adaptability*. Since $\mathbb{E}^t[Y|X]$ is unknown, $\lambda(\tilde{\mathcal{H}})$ is intractable in practical applications. By conveniently assuming that $\lambda(\tilde{\mathcal{H}})$ is negligible, it is possible to derive a Learning Objective 1 which minimizes $\tilde{\varepsilon}^s(g \circ \varphi) + \frac{1}{2} d(\tilde{\mathcal{H}} \Delta \tilde{\mathcal{H}})$. Indeed, minimizing $\mathbb{D}^{(s,t)}$ on $\varphi$ leads to learn representations $Z = \varphi(X)$ such that $\mathbb{P}^s(Z) = \mathbb{P}^t(Z)$, where such situation is sufficient to ensure that $d(\tilde{\mathcal{H}} \Delta \tilde{\mathcal{H}}) = 0$.

## 2.3 THE CURSE OF INVARIANCE IN DOMAIN ADAPTATION

Compressing representations is an easy way to achieve invariance. Assume $X^s \sim \mathcal{N}(0, 1)$, $X^t \sim \mathcal{U}(-1, 1)$ and $Y = \mathbf{1}_{X>0}$ then a trivial way to obtain representations invariance while separating source data is to consider $\varphi : x \mapsto \mathbf{1}_{x>0}$. The main drawback is a potential loss of information from the original feature space. To explain this phenomenon, we first introduce formally the notion of *Hypothesis space compression*:

**Definition 2** (Hypothesis space compression). *Let $(\varphi_1, \varphi_2) \in \Phi^2$, $\varphi_1$ compresses more $\mathcal{H}$ if $\tilde{\mathcal{H}}(\varphi_1) \subset \tilde{\mathcal{H}}(\varphi_2)$. By language abuse, we will say that $Z_1 = \varphi_1(X)$ is more compressed than $Z_2 = \varphi_2(X)$.*

Second, we build two possible solutions $(g_1, \varphi_1)$ and $(g_2, \varphi_2)$ of Learning Objective 1 by considering the following compositions $X \to_\varphi Z_1 \to_\psi Z_2 \to_g \hat{Y}$ such that $\varphi$ and $\psi \circ \varphi$ are both in $\Phi$ and $g$ and $g \circ \psi$ are both in $\mathcal{G}$. We set $g_1 = g \circ \psi$, $\varphi_1 = \varphi$ and $g_2 = g$, $\varphi_2 = \psi \circ \varphi$. These solutions are chosen to ensure that $g_1 \circ \varphi_1 = g_2 \circ \varphi_2$ and $Z_2 = \varphi_2(X)$ is more compressed than $Z_1 = \varphi_1(X)$. In the case when $\mathbb{D}^{(s,t)}$ is the Jensen divergence, the data processing inequality (see Lemma 4.6 in Zhao et al. (2019)) implies that $(g_2, \varphi_2)$ is a better optimum than $(g_1, \varphi_1)$ according to Learning Objective 1 since $\varepsilon^s(g_1 \circ \varphi_1) = \varepsilon^s(g_2 \circ \varphi_2)$ and $\mathbb{D}^{(s,t)}(\varphi_2) \leq \mathbb{D}^{(s,t)}(\varphi_1)$. To conclude, learning under objective 1 will favor compressed representations to enforce invariance. However, as noted in Johansson et al. (2019); Zhao et al. (2019); Liu et al. (2019), more compressed representations will degrade adaptability. Indeed,

$$\lambda(\tilde{\mathcal{H}}(\varphi_2)) = \inf_{h \in \tilde{\mathcal{H}}(\varphi_2)} \varepsilon^t(h) + \varepsilon^s(h) \leq \inf_{h \in \tilde{\mathcal{H}}(\varphi_1)} \varepsilon^t(h) + \varepsilon^s(h) = \lambda(\tilde{\mathcal{H}}(\varphi_1)) \tag{4}$$

since $\tilde{\mathcal{H}}(\varphi_1) \subset \tilde{\mathcal{H}}(\varphi_2)$. A question which remains not addressed in the literature (except in Johansson et al. (2019) under the Covariate Shift assumption) is:

*How compression may increase the risk of bad adaptability?*

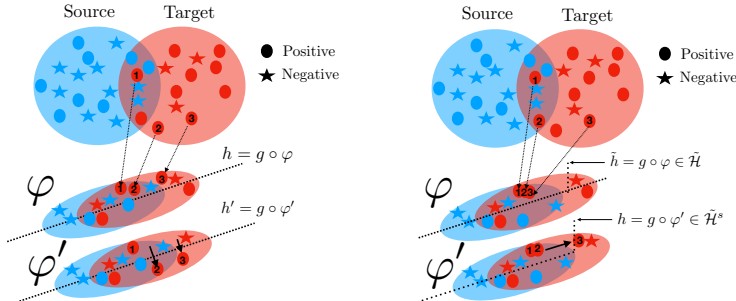

Figure 1: We fix $\varphi \in \Phi$ and we note $\mathcal{D}^d = \text{supp}(\mathbb{P}^d(X))$ and $\tilde{\mathcal{D}}^d = \text{supp}(\mathbb{P}^d(\varphi(X)))$ with $d = s, t$. Labels in domains are not necessary for the analysis but, for a better understanding, we draw $\varphi$ which separates well labels in the source domain since our analysis will consider representations trained by unsupervised domain adaptation. (**Left**) Illustration of the *source conservation* where $\varphi$ and $\varphi'$ are two source equivalent representations and $h = g \circ \varphi$ and $h' = g \circ \varphi'$ for a given $g \in \mathcal{G}$. Since $x_1 \in \mathcal{D}^s \cap \mathcal{D}^t$ we have $\varphi(x_1) = \varphi'(x_1)$. Note that $\varphi(x_3)$ may differ with $\varphi'(x_3)$ since $x_3 \notin \mathcal{D}^s$. This is also the case for $x_2$ even if $\varphi(x_2) \in \tilde{\mathcal{D}}^s$ since $x_2 \notin \mathcal{D}^s$. (**Right**) Illustration of the projected hypothesis for a given $\tilde{h} \in \tilde{\mathcal{H}}^s$ with $h = g \circ \varphi'$ and $g \in \mathcal{G}$. $\varphi$ embeds $x_1, x_2$ and $x_3$ closely in the representation space. The *source risk conservation* enforces to have $h = \tilde{h}$ on $\mathcal{D}^s$. Since $x_1$ and $x_2$ are in $\mathcal{D}^s \cap \mathcal{D}^t$, $\varphi$ and $\varphi'$ coincide on these points. $\varphi'$ modifies $x_3$ and changes its output for $g$. Then $\tilde{h}$ can not fit this switch of output for $x_3$ without changing outputs of both $x_1$ and $x_2$ illustrating the impact of *compression* of $\varphi$.

## 3 THEORETICAL ANALYSIS OF COMPRESSION

### 3.1 WORKING PLAN

Our strategy aims to compare a representation $\varphi$ with representations $\varphi'$ which act similarly on the source domain but differently on the target domain (*source equivalence*). First, by allowing to compute the minimal combined error on source equivalent representations $\varphi'$, we limit the risk of bad adaptability of a fixed representation. Through an inequality called *Target Compression*, we show this new control of adaptability comes at the cost of a new risk which embodies the *Compression*. This is done by studying the *projection* of an hypothesis defined as its best approximation from representations in the target domain among those that conserve the source domain error (*source conservation*). Second, we rely on the $\tilde{\mathcal{H}}\Delta\tilde{\mathcal{H}}$ inequality (Ben-David et al. (2007; 2010)) for stating a new bound of the target risk (*Invariant and Compressed Representations* inequality). Finally, we expose the particular interest of this bound: it allows a better control on adaptability by considering compression. This is formulated by the *Invariance - Compression trade-off*.

### 3.2 NOTATIONS AND DEFINITIONS

In the following, we fix a given $\varphi \in \Phi$. We restrict our analysis to the case of $\ell : (y, y') \in \mathcal{Y}^2 \mapsto (y - y')^2$ as a convenient choice for manipulating properly $\tilde{f}(Z) = \mathbb{E}[Y|Z]$. Note that in the binary classification case ($\mathcal{Y} = \{0, 1\}$), $\ell$ is the accuracy. We define the set of representations which act similarly on the source domain but differently in the target domain:

**Definition 3** (Source equivalence). *We introduce the set of source equivalent representations $\tilde{\Phi}^s = \{\varphi' \in \Phi, \varphi' = \varphi \text{ on } \mathcal{D}^s\}$ (where $\mathcal{D}^s = \text{supp}(\mathbb{P}^s(X))$) and the source equivalent hypothesis space $\tilde{\mathcal{H}}^s = \{g \circ \varphi' : g \in \mathcal{G}, \varphi' \in \tilde{\Phi}^s\}$. Two elements of $\tilde{\Phi}^s$ are said source equivalent.*

The source equivalent hypothesis space $\tilde{\mathcal{H}}^s$ is an extension of $\tilde{\mathcal{H}}$ adding hypothesis built using representations which are source equivalent to $\varphi$. For a given $h \in \tilde{\mathcal{H}}^s$, we exhibit an interesting subset of $\tilde{\mathcal{H}}$ considering hypothesis with the same source error than $h$.

**Definition 4** (Source conservation). *For a given $h \in \tilde{\mathcal{H}}^s$, we note $\tilde{\mathcal{H}}_h = \{h' \in \tilde{\mathcal{H}} : \varepsilon^s(h, h') = 0\}$.*

It is worth noting that for all $h \in \tilde{\mathcal{H}}^s$, $\tilde{\mathcal{H}}_h$ is not empty (see Appendix A for details). We are now ready to introduce the projection of an hypothesis $h \in \tilde{\mathcal{H}}^s$ and to this purpose, we note $\varphi' \in \tilde{\Phi}^s$ and $g' \in \mathcal{G}$ such that $h = g' \circ \varphi'$. Intuitively, the projection of $h$ is defined as its best approximation from $\varphi$ in the target domain among those which verify the source conservation *i.e.* from hypothesis $g \circ \varphi$ with $g \in \mathcal{G}$ such that $\varepsilon^s(g \circ \varphi, h) = 0$.

**Definition 5** (Projection). $\pi : \tilde{\mathcal{H}}^s \to \tilde{\mathcal{H}}$ *is defined as follows:* $\pi(h) = \arg\min_{h' \in \tilde{\mathcal{H}}_h} \varepsilon^t(h, h')$ *for* $h \in \tilde{\mathcal{H}}^s$. $\pi(h)$ *is called the projected hypothesis. If there is no uniqueness of the minimum, any solution works.*

Illustrations of introduced definitions are provided in Figure 1. If $h = g \circ \varphi \in \tilde{\mathcal{H}}$, the projection aims to map $h$ to $g' \circ \varphi$ with $g' = g$ both in source and target $L^2$ norms. If, $h = g' \circ \varphi'$ with $\varphi' \in \tilde{\Phi}$, the projection aims to map $h$ to $g \circ \varphi$ such that $g = g'$ in source $L^2$ norm while $g \circ \varphi$ achieves a minimal distance with $g' \circ \varphi'$ in target $L^2$ norm. We can observe that $\forall h \in \mathcal{H}^s, \pi \circ \pi(h) = \pi(h)$ in both source and target $L^2$ norms hence the (abusive) analogy with projection.

For ease of reading, we remind the notations and definitions in appendix B.

### 3.3 OUR CONTRIBUTION

In the following, we fix a given $\mathcal{H}_\circ$ such that $\tilde{\mathcal{H}} \subset \mathcal{H}_\circ \subset \tilde{\mathcal{H}}^s$. We are now ready to state a new bound of the target risk:

**Inequality 2** (Target Compression). *Given $g \in \mathcal{G}$ and noting $\tilde{h} = \pi(h)$, we have:*

$$\varepsilon^t(g \circ \varphi) \leq \varepsilon^t(g \circ \varphi, \tilde{f}^s \circ \varphi) + \underbrace{\sup_{h \in \mathcal{H}_\circ} \varepsilon^t(h, \tilde{h})}_{\tilde{\gamma}(\mathcal{H}_\circ): \ Compression} + \underbrace{\inf_{h \in \mathcal{H}_\circ} \{\varepsilon^t(h) + \varepsilon^t(\tilde{h}, \tilde{f}^s \circ \varphi)\}}_{\tilde{\lambda}^t(\mathcal{H}_\circ): \ Adaptability} \quad (5)$$

*Proof.* See Appendix A. $\square$

This inequality introduces two interesting terms $\tilde{\gamma}(\mathcal{H}_\circ)$ and $\tilde{\lambda}^t(\mathcal{H}_\circ)$ with opposite roles; the former involves a supremum while the latter involves an infremum on $\mathcal{H}_\circ$. The term $\tilde{\lambda}^t(\mathcal{H}_\circ)$ is very similar to the adaptability $\lambda(\tilde{\mathcal{H}})$ from Inequality 1 and will be further investigated. Before, we focus on the term $\tilde{\gamma}(\mathcal{H}_\circ)$ which reflects the possibility of finding hypotheses in $\mathcal{H}_\circ$ that can not be well approximated from target representations. The more information is lost from raw features, the higher the risk, hence its name *Compression*. This can be formally stated as follows:

**Theorem 1** ($\gamma$ embodies the risk of compression). *Let $(\varphi_1, \varphi_2) \in \Phi^2$ two source equivalent source representations. If $\varphi_1$ is more compressed than $\varphi_2$, then $\gamma(\varphi_1, \mathcal{H}_\circ) \geq \gamma(\varphi_2, \mathcal{H}_\circ)$.*

*Proof.* If $\varphi_1$ is more compressed than $\varphi_2$, there is $\tilde{\mathcal{H}}(\varphi_1) \subset \tilde{\mathcal{H}}(\varphi_2) \subset \tilde{\mathcal{H}}^s$ then $\forall h \in \tilde{\mathcal{H}}^s, \tilde{\mathcal{H}}_h(\varphi_1) \subset \tilde{\mathcal{H}}_h(\varphi_2)$. The projection involves an infremum then $\varepsilon^t(h, \tilde{h}(\varphi_2)) \leq \varepsilon^t(h, \tilde{h}(\varphi_1))$ where $\tilde{h}(\varphi) = \arg\min_{h' \in \tilde{\mathcal{H}}_h(\varphi)} \varepsilon^t(h, h')$. The supremum on $h \in \mathcal{H}_\circ$ conserves the inequality. $\square$

We now focus our attention on $\tilde{\lambda}^t(\mathcal{H}_\circ)$. It differs with $\lambda(\tilde{\mathcal{H}})$ in two ways. First, the source labelling function is involved only with the intermediate $\tilde{f}^s(Z) = \mathbb{E}^s[Y|Z]$. Second, the risk associated with it ($\varepsilon^t(\tilde{h}, \tilde{f}^s \circ \varphi)$) involves a target expectation. In order to obtain a comparable bound with Inequality 1, we state a new bound of the target risk which is our main contribution:

**Inequality 3** (Invariant and Compressed Representations). *Given $g \in \mathcal{G}$ and noting $\tilde{h} = \pi(h)$, $\beta = 2\inf_{h \in \tilde{\mathcal{H}}} \varepsilon^s(h) + \varepsilon^t(h, \tilde{f}^s)$, we have:*

$$\varepsilon^t(g \circ \varphi) \leq \varepsilon^s(g \circ \varphi) + d(\tilde{\mathcal{H}}\Delta\tilde{\mathcal{H}}) + \beta + \underbrace{\sup_{h \in \mathcal{H}_\circ} \varepsilon^t(h, \tilde{h})}_{\tilde{\gamma}(\mathcal{H}_\circ): \ Compression} + \underbrace{\inf_{h \in \mathcal{H}_\circ} \varepsilon^t(h) + \varepsilon^s(h)}_{\tilde{\lambda}(\mathcal{H}_\circ): \ Adaptability} \quad (6)$$

*Proof.* See AppendixA. The main idea is to bound $\varepsilon^t(h, \tilde{f}^s \circ \varphi)$ and $\varepsilon^t(\tilde{h}, \tilde{f}^s)$ using $\tilde{\mathcal{H}}\Delta\tilde{\mathcal{H}}$ inequality. This introduces respectively $\varepsilon^s(h, \tilde{f}^s \circ \varphi)$ and $\varepsilon^s(\tilde{h}, \tilde{f}^s \circ \varphi)$ with residual terms $\frac{1}{2}d(\tilde{\mathcal{H}}\Delta\tilde{\mathcal{H}}) + \frac{1}{2}\beta$ (two times). We use the source risk conservation to bound $\varepsilon^s(\tilde{h}, \tilde{f}^s \circ \varphi) \leq \varepsilon^s(h, \tilde{f}^s \circ \varphi)$. Then we bound $\varepsilon^s(h, \tilde{f}^s \circ \varphi) \leq \varepsilon^s(h)$ leveraging the Pythagorean theorem since $\ell$ is the $L^2$ norm. $\qquad\square$

We are now ready to show the better control of the adaptability of the representation $\varphi$:

**Theorem 2** (Better control of adaptability). $\tilde{\lambda}(\mathcal{H}_\circ) \leq \lambda(\tilde{\mathcal{H}})$.

The proof is given by the infremum on $\mathcal{H}_\circ \supset \tilde{\mathcal{H}}$. Compression $\tilde{\gamma}(\mathcal{H}_\circ)$ and adaptability $\tilde{\lambda}(\mathcal{H}_\circ)$ have an opposite behavior. This reflects an interesting trade-off:

**Theorem 3** (Invariance and Compression trade-off). *With $\mathcal{H}_\circ^1$ and $\mathcal{H}_\circ^2$ such that $\tilde{\mathcal{H}} \subset \mathcal{H}_\circ^1 \subset \mathcal{H}_\circ^2 \subset \tilde{\mathcal{H}}^s$, there is $\tilde{\lambda}(\mathcal{H}_\circ^1) \geq \tilde{\lambda}(\mathcal{H}_\circ^2)$ and $\tilde{\gamma}(\mathcal{H}_\circ^1) \leq \tilde{\gamma}(\mathcal{H}_\circ^2)$.*

The proof is obtained by observing that $\mathcal{H}_\circ^1 \subset \mathcal{H}_\circ^2$. Even if invariance may have a bad impact on adaptability, our bound shows it is possible to have a better control of the adaptability while conserving the same degree of invariance. This control comes at the cost of an additional term involving compression of representations. The trade-off between invariance and compression emphasizes than the more control there is over adaptability, the higher the risk associated with compression. Furthermore, this new bound has a particular interest in providing theoretical support for the construction of new learning objectives. The difficulty in DA comes from in the fact that the adaptability is not trainable since it uses the labelling function in the target domain. Our study shows that better control of adaptability (*i.e.* making the adaptation more reliable) is possible. The two additional terms $\beta$ and $\tilde{\gamma}(\mathcal{H}_\circ)$ can indeed be trained in an Unsupervised Domain Adaptation setting[1].

## 4 EXPERIMENTS

We are now conducting an experimental analysis on a widely used DA benchmark of digits recognition (MNIST, USPS). We study two tasks M→U and U→M under the Learning Objective 1. The representation $\varphi$ is a convolutional neural network with a LeNet architecture (LeCun et al. (1998)). We set the dimension of representations to 50 and $\mathcal{G}$ is the set of linear classifiers from the representation space to classes (with a softmax layer on top) and the Domain Adversarial Neural Network (DANN Ganin & Lempitsky (2015)) for $\mathbb{D}^{(s,t)}$. We use the implementation from Long et al. (2018) for good reproducibility. Models are trained during 30 epochs and we note $\hat{\varphi}$ the learned representation. In the following, we note $\mathbb{CE}$ the cross-entropy.

We suggest to study the trade-off between compression and invariance with the family:

$$\mathcal{H}_\circ^\eta = \{g \circ \varphi' \in \tilde{\mathcal{H}}^s : \mathbb{E}^t ||\varphi'(X) - \varphi(X)|| \leq \eta \mathbb{E}^t ||\varphi(X)||\}, \quad \eta > 0 \tag{7}$$

We interpret this choice as a method for testing robustness of the learned representation to adversarial attacks in the target domain.

**Enforcing** $h \in \mathcal{H}_\circ^\eta$. Enforcing $h \in \mathcal{H}_\circ^\eta$ is challenging in practice. We suggest to rely on a penalization $\mathcal{L}_\circ$ to promote $\varphi \approx \hat{\varphi}$ (in the sense of $\mathbb{P}^s$) and $||\varphi - \hat{\varphi}|| \lesssim \eta \cdot ||\hat{\varphi}||$ (in the sense of $\mathbb{P}^t$). We define the penalization $\mathcal{L}_\circ$ as follows:

$$\mathcal{L}_\circ(\varphi) = \frac{\mathbb{E}^s ||\varphi(X) - \hat{\varphi}(X)||}{\mathbb{E}^s ||\hat{\varphi}(X)||} + \text{ReLU}\left(\frac{\mathbb{E}^t ||\varphi(X) - \hat{\varphi}(X)||}{\mathbb{E}^t ||\hat{\varphi}(X)||} - \eta\right) \tag{8}$$

**Source conservation.** For a given $h$, the projected hypothesis is defined as $\tilde{h} = \tilde{g} \circ \hat{\varphi}$ where $\tilde{g}$ is obtained by minimizing on $g$:

$$\Pi(g, h) = \lambda_\pi \cdot \mathbb{CE}^s(g \circ \hat{\varphi}(X), h(X)) + (1 - \lambda_\pi) \cdot \mathbb{CE}^t(g \circ \hat{\varphi}(X), h(X)) \tag{9}$$

with $\lambda_\pi = 0.9$ for enforcing the source conservation.

---

[1] The trainability of $\beta$ is shown in Appendix A

**Estimation of $\tilde{\lambda}(\mathcal{H}_\circ)$.** We suggest to minimize on $(g, \varphi) \in \mathcal{H}$ the following loss:

$$\Lambda(g, \varphi) = \mathbb{C}\mathbb{E}^t(g \circ \varphi(X), Y) + \mathbb{C}\mathbb{E}^s(g \circ \varphi(X), Y) + \lambda_\circ \cdot \mathcal{L}_\circ(\varphi) \tag{10}$$

with $\lambda_\circ = 10^2$. $\mathbb{C}\mathbb{E}^t(g \circ \varphi(X), Y)$ and $\mathbb{C}\mathbb{E}^s(g \circ \varphi(X), Y)$ are respectively two proxies of $\varepsilon^t(h)$ and $\varepsilon^s(h)$. $\mathcal{L}_\circ(\varphi)$ promotes $h \in \mathcal{H}_\circ^\eta$.

**Estimation of $\tilde{\gamma}(\mathcal{H}_\circ)$.** We follow an adversarial training procedure alternating between

1. maximizing an objective $\Gamma(g, \varphi)$ on $h = g \circ \varphi \in \mathcal{H}_\circ$ (sup step)
2. computing the projected hypothesis (inf step)

During experimentation, we have observed that simply setting $\Gamma(g, \varphi) = -\Pi(\hat{g} \circ \hat{\varphi}, g \circ \varphi) + \lambda_\circ \mathcal{L}_\circ(\varphi)$ (where $\hat{g} \circ \hat{\varphi}$ is an estimation of the projected hypothesis) leads to learn noisy labels. We suggest to prevent such pathological solutions by enforcing a low entropy adding to $\Gamma$ a regularizer $\lambda_{\text{ent}} \cdot \mathbb{C}\mathbb{E}(h(X), \hat{Y})$ with $(\hat{Y} = y) = \mathbf{1}(h(X) = y)$ with $\lambda_{\text{ent}} = -1$. Then, we set:

$$\Gamma(g, \varphi) = -\Pi(\hat{g} \circ \hat{\varphi}, g \circ \varphi) + \lambda_\circ \mathcal{L}_\circ(\varphi) + \lambda_{\text{ent}} \cdot \mathbb{C}\mathbb{E}(h(X), \hat{Y}) \tag{11}$$

Furthermore, for a fair comparison, we correct $\gamma$ by comparing it with a baseline. This baseline is allowed to train representations to fit the adversary.

**Analysis** We build new target domains modifying the label distribution which is known to badly impact $\lambda(\tilde{\mathcal{H}})$ (Johansson et al. (2019); Zhao et al. (2019)). For a given target dataset $\mathcal{D}$ with classes balanced, we define $\mathcal{D}_1 \subset ... \subset \mathcal{D}_6$ as follows: for $1 \leq i \leq 6$, $\mathcal{D}_i$ is obtained by removing half of digit $< i$ from $\mathcal{D}_{i-1}$ and setting $\mathcal{D}_0 = \mathcal{D}$. Main results are reported in Figure 2. Although estimator are trained using cross-entropy, the results are reported computing the accuracy on a test set in order to be consistent with our theoretical analysis.

We can observe that stronger the adversary is ($\alpha \to 1$) (and the more target shift ($i \to 6$ for $\mathcal{D}_i$)), the more significant the difference between $\lambda(\tilde{\mathcal{H}})$ and $\tilde{\lambda}(\mathcal{H}_\circ)$. This demonstrates a better control over adaptability especially when IR fails to perform adaptation. By playing with the value of $\eta$, we see that the better control on adaptability is paid to the cost of a higher $\gamma$. In our example, the risk of compression is estimated to a potential drop of performances from $\sim$12.5% to $\sim$27.5% of accuracy in the target domain while the gain on adaptability is marginal by reducing it of $\sim$4%. To conclude, a better control on adaptability is obtained with a pessimistic estimation of the risk of compression.

## 5 WEIGHTS FOR INVARIANCE RELAXATION

The fact that compression and invariance are conflicting objectives supports the point from Wu et al. (2019) which claims that invariance is too strict a constraint. In this section, we show that invariance can be relaxed leveraging *Weighted Representations*. More specifically, we show that taking into account the risk of compression leads to introduce an adaptability term which does not depend on learned weights. Then, weighting representations can reduce the discrepancy between the source and the target distributions without impacting the adaptability. We introduce the set of weights $\mathcal{W}$, a subset of functions $w : \mathcal{X} \to \mathbb{R}^+$ such that $\mathbb{E}^s[w(X)] = 1$ and we note for $w \in \mathcal{W}, \mathbb{P}^{w \cdot s}(X) := w(X)\mathbb{P}^s(X)$. The condition $\mathbb{E}^s[w(X)] = 1$ ensures that $\mathbb{P}^{w \cdot s}(X)$ is a distribution. We state a similar result than Inequality 1.

**Inequality 4** (Weighted version of Inequality1). *For a given $\varphi \in \Phi$ and $g \in \tilde{\mathcal{H}}$,*

$$\varepsilon^t(g \circ \varphi) \leq \varepsilon^{w \cdot s}(g \circ \varphi) + \frac{1}{2}d_w(\tilde{\mathcal{H}}\Delta\tilde{\mathcal{H}}) + \lambda_w(\tilde{\mathcal{H}}) \tag{12}$$

*with $d_w(\tilde{\mathcal{H}}\Delta\tilde{\mathcal{H}}) = \sup_{(h, h') \in \tilde{\mathcal{H}}} |\varepsilon^{w \cdot s}(h, h') - \varepsilon^t(h, h')|$ and $\lambda_w = \inf_{h \in \tilde{\mathcal{H}}} \varepsilon^t(h) + \varepsilon^{w \cdot s}(h)$.*

Interestingly, $w$ and $\varphi$ play similar role in inequality 4 since they are actionable parameters to minimize $\varepsilon^{w \cdot s}(g \circ \varphi) + \frac{1}{2}d_w(\tilde{\mathcal{H}}\Delta\tilde{\mathcal{H}})$ while we can not quantify their impact on $\lambda_w(\tilde{\mathcal{H}})$. We show that taking advantage of compression leads to an adaptability which does not depend on weights. This needs to introduce some definitions:

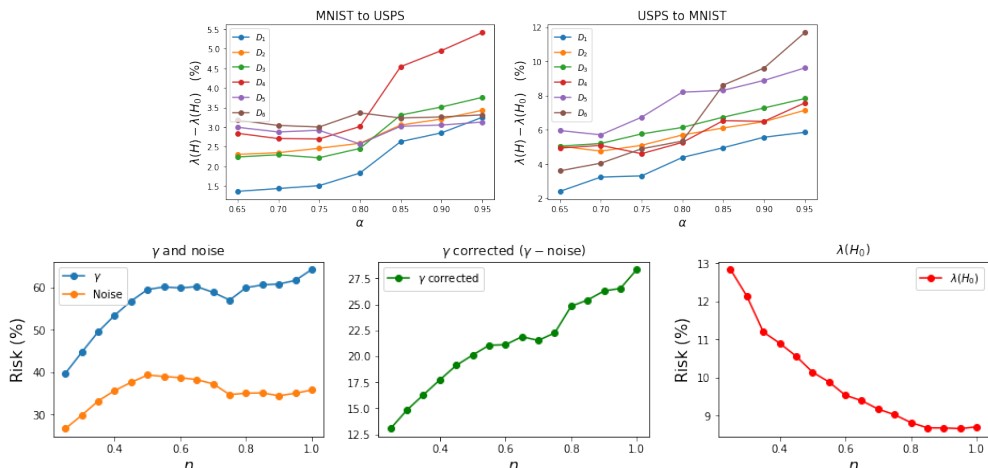

Figure 2: **Up:** We report the $\Delta(\lambda) = \lambda(\tilde{\mathcal{H}}) - \tilde{\lambda}(\mathcal{H}_\circ^\eta)$ with $\eta = 1$. $\Delta(\lambda) \geq 0$ which supports Theorem 2. More the adversary is strong in LO 1 ($\alpha \to 1$), more $\Delta(\lambda)$ increases (similar analysis for $\mathcal{D}_1$ to $\mathcal{D}_5$). **Down:** Illustration of the trade-off from Theorem 3 in the particular case of $\mathcal{D}_4$ with $\alpha = 0.75$ on U $\to$ M. Both $\gamma$ and its correction are reported. When $\eta$ increases, the risk of compression increases while the adaptability assumption is better controlled. *Better in color.*

**Definition 6** (Strong source conservation). *For a given $h \in \tilde{\mathcal{H}}^s$, we note $\tilde{\mathbf{H}}_h = \{h' \in \tilde{\mathcal{H}} : h = h'$ on $\mathcal{D}^s\}$ where $\mathcal{D}^s = \text{supp}(\mathbb{P}^t(X))$.*

**Definition 7** (Strong projection). $\mathbf{\Pi} : \tilde{\mathcal{H}}^s \to \tilde{\mathcal{H}}$ *is defined as follows* $\pi(h) = \arg\min_{h' \in \tilde{\mathbf{H}}_h} \varepsilon^t(h, h')$ *for $h \in \tilde{\mathcal{H}}^s$. If there is no uniqueness of the minimum, any solution works.*

We are now ready to state the role of weights as a relaxed invariance for a lower compression:

**Inequality 5** (Weighting Invariant and Compressed Representations). *Given $g \in \mathcal{G}$ and noting $\mathbf{h} = \mathbf{\Pi}(h)$, $\beta_w = 2\inf_{h \in \tilde{\mathcal{H}}} \varepsilon^{w \cdot s}(h) + \varepsilon^t(h, \tilde{f}^s)$, we have:*

$$\varepsilon^t(g \circ \varphi) \leq \inf_{w \in \mathcal{W}} \left\{ \varepsilon^{w \cdot s}(g \circ \varphi) + \frac{1}{2} d_w(\tilde{\mathcal{H}} \Delta \tilde{\mathcal{H}}) + \frac{1}{2}\beta_w \right\} + \sup_{h \in \mathcal{H}_\circ} \varepsilon^t(h, \mathbf{h}) + \inf_{h \in \mathcal{H}_\circ} \varepsilon^t(h) + \varepsilon^t(\mathbf{h}, \tilde{f}^s \circ \varphi)$$

*Proof.* This is a straightforward extension of inequality 2 applied to $\mathbb{P}^{w \cdot s}$. Furthermore, $\inf_{h \in \mathcal{H}_\circ} \varepsilon^t(h) + \varepsilon^t(\mathbf{h}, \tilde{f}^s \circ \varphi)$ does not depend on $w$ due to **strong** risk conservation (risk conservation would have been dependent on $w$ through the constraint $\varepsilon^{w \cdot s}(h, \tilde{h}) = 0$). $\square$

Adding weights exposes to the risk of increasing the variance of estimators when considering finite sampling analysis (see Cortes et al. (2010) for an extensive theoretical analysis of Importance Sampling). In inequality 5, this can be controlled with a good choice of $\mathcal{W}$ *e.g.* by considering weights $w \leq M$ for a given $M > 0$.

## 6 RELATED WORK

Generalization bounds for DA have been the object of an extensive literature for both Importance Sampling Cortes et al. (2010) and Invariant Representations Ben-David et al. (2007; 2010); Mansour et al. (2009). It has attracted a lot of attention in order to understand failure cases observed when using IR (Zhao et al. (2019); Johansson et al. (2019)). The theoretical analysis in Johansson et al. (2019) is the work closest to ours. They introduce a *loss information* metric which quantifies the risk of using non-invertible representations. The present work differs in three ways. First, our work follows the formalism introduced in Ben-David et al. (2007) which is widely adopted in the community. Second, contrary to the *information loss*, our theory provides a tractable risk of compressing representations. Finally, we do not rely on the *Covariate Shift* assumption which gives our work a broader range of applications.

## 7    CONCLUSION

By addressing domain generalization under the perspective of compression, we have bounded the target risk by a sum of trainable terms ($\varepsilon^s(g \circ \varphi) + d(\tilde{\mathcal{H}}\Delta\tilde{\mathcal{H}}) + \beta + \tilde{\gamma}(\mathcal{H}_\circ)$) and a term $\tilde{\lambda}(\mathcal{H}_\circ)$ which embeds a new adaptability criterion of representations. We have shown that $\tilde{\lambda}(\mathcal{H}_\circ)$ can be better controlled by enriching the hypothesis class $\mathcal{H}_\circ$. This control comes at a cost of a higher risk of compression underlying a trade-off between invariance and compression when learning IR for DA. Through an empirical study on a standard benchmark, we have exhibited the better control of adaptability. Furthermore, we have shown how our framework can be applied for studying adaptation robustness to adversarial attacks. Finally, as an attempt to obtain representation distributions invariance with a lower compression, we have emphasized the role of weighted representations for relaxing the constraint of invariance.

Achieving representation invariance can reduce the generalization gap between two domains. However, all invariances will not have the same guarantees for better generalization. Our work shows that we will look for representations which preserve the best the information in the original target features spaces. We leave both the design on new Domain Adaptation methods which incorporate such consideration and relevant design of $\mathcal{H}_\circ$ as future works.

**Promising directions**   (1) Our analysis is restricted to the comparison among source equivalent representations of a given representation $\varphi \in \Phi$. Then, for two given representations $\varphi_1$ and $\varphi_2$ of $\Phi$ which does not verify the source equivalence, our analysis does not allow to state if $\varphi_1$ presents, or not, better guarantees on the success of the adaptation. Understanding how the source equivalence can be relaxed can provide to the present work a broader range of applications. (2) Obtaining a robust estimation of $\tilde{\gamma}(\mathcal{H}_\circ)$ by following an adversarial training procedure appeared to be challenging in practice. Finding fast and robust estimations of $\tilde{\gamma}(\mathcal{H}_\circ)$ (or a proxy of it) may help to derive new learning objectives or policy for model selection. For instance, *Information Theory* gives better suited devices for modelling the phenomenon of compression. (3) A strategy may to enforce invariance directly in the target domain by transporting source samples. Formally, assuming $\mathcal{D}^s \cap \mathcal{D}^t = \emptyset$, we set $\varphi$ such that $\varphi(X) = X$ on $\mathcal{D}^t$ and $\varphi(\mathcal{D}^s) \subset \mathcal{D}^t$. If $\varphi$ is fixed in the source domain (*i.e.*  $\varphi_{|\mathcal{D}^s}$ is fixed) the optimal representation from the compression perspective is $\varphi_{|\mathcal{D}^t}(X) = X$. This consideration is connected with two recent trends which perform adaptation by generating cross domain samples (Sankaranarayanan et al. (2018); Hoffman et al. (2018); Bousmalis et al. (2016)) or by finding *Optimal Transport* between the source and the target domains (Courty et al. (2016; 2017)). (4) The *Covariate Shift* assumption can be stated as $\mathcal{H}_\circ = \{h \in \mathcal{H}, \varepsilon^s(h, \hat{h}) = 0 \text{ with } \hat{h} = \arg\min_{h \in \mathcal{H}} \varepsilon^s(h)\}$. Such $\mathcal{H}_\circ$ may provide a new framework for addressing the case of non-overlapping supports in Covariate Shift adaptation.

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

## A    MAIN PROOFS

**Proposition 1** (The source conservation hypothesis is not empty)**.** *For a given $\varphi \in \Phi$ and $h \in \tilde{\mathcal{H}}^s$, $\tilde{\mathcal{H}}_h$ is not empty.*

*Proof.* If $h \in \tilde{\mathcal{H}}^s$, it exists $g \in \mathcal{G}$ and $\varphi' \in \Phi$ such that $h = g \circ \varphi'$. We set $h' = g \circ \varphi \in \tilde{\mathcal{H}}$, there is $\varepsilon^s(h, h') = \varepsilon^s(g \circ \varphi', g \circ \varphi) = \varepsilon^s(g \circ \varphi, g \circ \varphi) = 0$ since $\varphi = \varphi'$ on $\mathcal{D}^s$ then $h' \in \tilde{\mathcal{H}}_h$.    $\square$

**Inequality 6** (Target Compression)**.** *Given $\varphi \in \Phi$, $g \in \mathcal{G}$ and noting $\tilde{h} = \pi(h)$, there is for any $\mathcal{H}_\circ$ such that $\tilde{\mathcal{H}} \subset \mathcal{H}_\circ \subset \tilde{\mathcal{H}}^s$:*

$$\varepsilon^t(g \circ \varphi) \leq \varepsilon^t(g \circ \varphi, \tilde{f}^s \circ \varphi) + \sup_{h \in \mathcal{H}_\circ} \varepsilon^t(h, \tilde{h}) + \inf_{h \in \mathcal{H}_\circ} \{\varepsilon^t(h) + \varepsilon^t(\tilde{h}, \tilde{f}^s \circ \varphi)\} \qquad (13)$$

*Proof.* Let $\varphi \in \Phi$, $g \in \mathcal{G}$ and $h \in \mathcal{H}_\circ$:

$$\varepsilon^t(g \circ \varphi) \leq \varepsilon^t(\tilde{f}^s \circ \varphi) + \varepsilon^t(\tilde{f}^s \circ \varphi, g \circ \varphi) \qquad (14)$$

$$\leq \varepsilon^t(h) + \varepsilon^t(h, \tilde{f}^s \circ \varphi) + \varepsilon^t(\tilde{f}^s \circ \varphi, g \circ \varphi) \qquad (15)$$

$$\leq \varepsilon^t(h) + \varepsilon^t(h, \tilde{h}) + \varepsilon^t(\tilde{h}, \tilde{f}^s \circ \varphi) + \varepsilon^t(\tilde{f}^s \circ \varphi, g \circ \varphi) \qquad (16)$$

$$\leq \varepsilon^t(\tilde{f}^s \circ \varphi, g \circ \varphi) + \varepsilon^t(h, \tilde{h}) + \varepsilon^t(h) + \varepsilon^t(\tilde{h}, \tilde{f}^s \circ \varphi) \qquad (17)$$

This is a succession of triangular inequalities which involve respectively $\tilde{f}^s \circ \varphi$, $h$ and $\tilde{h}$. The last inequality is a convenient arrangement of terms. We bound $\varepsilon^t(h, \tilde{h})$ by its supremal value for $h \in \mathcal{H}_\circ$, then $\varepsilon^t(g \circ \varphi) \leq \varepsilon^t(\tilde{f}^s \circ \varphi, g \circ \varphi) + \sup_{h \in \mathcal{H}_\circ} \{\varepsilon^t(h, \tilde{h})\} + \varepsilon^t(h) + \varepsilon^t(\tilde{h}, \tilde{f}^s \circ \varphi)$. Since this holds for all $h \in \mathcal{H}_\circ$, we bound $\varepsilon^t(g \circ \varphi)$ by the infremum for $h \in \mathcal{H}_\circ$.    $\square$

**Inequality 7** (Invariant and Compressed Representations)**.** *Given $\varphi \in \Phi$, $g \in \mathcal{G}$ and noting $\tilde{h} = \pi(h)$, $\beta = 2 \inf_{h \in \tilde{\mathcal{H}}} \varepsilon^s(h) + \varepsilon^t(h, \tilde{f}^s)$, there is for any $\mathcal{H}_\circ$ such that $\tilde{\mathcal{H}} \subset \mathcal{H}_\circ \subset \tilde{\mathcal{H}}^s$:*

$$\varepsilon^t(g \circ \varphi) \leq \varepsilon^s(g \circ \varphi) + d(\tilde{\mathcal{H}} \Delta \tilde{\mathcal{H}}) + \beta + \underbrace{\sup_{h \in \mathcal{H}_\circ} \varepsilon^t(h, \tilde{h})}_{\tilde{\gamma}(\mathcal{H}_\circ):\ Compression} + \underbrace{\inf_{h \in \mathcal{H}_\circ} \varepsilon^t(h) + \varepsilon^s(h)}_{\tilde{\lambda}(\mathcal{H}_\circ):\ Adaptability} \qquad (18)$$

*Proof.* First of all, we recall the $\tilde{\mathcal{H}} \Delta \tilde{\mathcal{H}}$ inequality from Ben-David et al. (2007; 2010), for any $f^s$ and $f^t$, there is:

$$\forall h \in \tilde{\mathcal{H}}, \varepsilon^t(h, f^t) \leq \varepsilon^s(h, f^s) + \frac{1}{2} d(\tilde{\mathcal{H}} \Delta \tilde{\mathcal{H}}) + \inf_{h \in \tilde{\mathcal{H}}} \varepsilon^t(h, f^t) + \varepsilon^s(h, f^s) \qquad (19)$$

A particular case of that inequality applied to $f^s = f^t = \tilde{f}^s \circ \varphi$ leads to bound:

$$\varepsilon^t(g \circ \varphi, \tilde{f}^s \circ \varphi) \leq \varepsilon^s(g \circ \varphi, \tilde{f}^s \circ \varphi) + \frac{1}{2} d(\tilde{\mathcal{H}} \Delta \tilde{\mathcal{H}}) + \inf_{h \in \tilde{\mathcal{H}}} \varepsilon^s(h, \tilde{f}^s \circ \varphi) + \varepsilon^t(h, \tilde{f}^s \circ \varphi)$$

$$\leq \varepsilon^s(g \circ \varphi) + \frac{1}{2} d(\tilde{\mathcal{H}} \Delta \tilde{\mathcal{H}}) + \inf_{h \in \tilde{\mathcal{H}}} \varepsilon^s(h) + \varepsilon^t(h, \tilde{f}^s \circ \varphi) \qquad (20)$$

where bounding $\varepsilon^s(g \circ \varphi, \tilde{f}^s \circ \varphi) \leq \varepsilon^s(g \circ \varphi)$ is a direct property of $\ell(y, y') = (y - y')^2$ and $\tilde{f}^s(Z) = \mathbb{E}^s[Y|Z]$: $\varepsilon^s(g \circ \varphi) = \varepsilon^s(g \circ \varphi, \tilde{f}^s \circ \varphi) + \varepsilon^s(\tilde{f}^s \circ \varphi)$. Note that, it is not possible to do so for $\varepsilon^t(g \circ \varphi, \tilde{f}^s \circ \varphi)$ since the expectation is computed in the target domain. We define $\beta = 2 \inf_{h \in \tilde{\mathcal{H}}} \varepsilon^s(h) + \varepsilon^t(h, \tilde{f}^s \circ \varphi)$. We apply a second time $\tilde{\mathcal{H}} \Delta \tilde{\mathcal{H}}$ inequality, there is for any $h \in \mathcal{H}_\circ$:

$$\varepsilon^t(\tilde{h}, \tilde{f}^s \circ \varphi) \leq \varepsilon^s(\tilde{h}, \tilde{f}^s \circ \varphi) + \frac{1}{2} d(\tilde{\mathcal{H}} \Delta \tilde{\mathcal{H}}) + \inf_{h \in \tilde{\mathcal{H}}} \varepsilon^t(h, \tilde{f}^s \circ \varphi) + \varepsilon^s(h, \tilde{f}^s \circ \varphi)$$

$$\leq \varepsilon^s(\tilde{h}) + \frac{1}{2} d(\tilde{\mathcal{H}} \Delta \tilde{\mathcal{H}}) + \inf_{h \in \tilde{\mathcal{H}}} \varepsilon^t(h, \tilde{f}^s) + \varepsilon^s(h) \qquad (21)$$

which is licit since $\tilde{h} \in \tilde{\mathcal{H}}$. Relying on the projected hypothesis $\tilde{h}$ allows us to preserve the invariance: even if $h \in \mathcal{H}_\circ$, $d(\tilde{\mathcal{H}}\Delta\tilde{\mathcal{H}})$ is involved, not $d(\mathcal{H}_\circ\Delta\mathcal{H}_\circ)$. Then, we use the property of source risk conservation $h \in \mathcal{H}$ which enforces $\varepsilon^s(\tilde{h}, h) = 0$ for noting that $\varepsilon^s(\tilde{h}) \leq \varepsilon^s(h) + \varepsilon^s(h, \tilde{h}) = \varepsilon^s(h)$ then:

$$\varepsilon^t(\tilde{h}, \tilde{f}^s \circ \varphi) \leq \varepsilon^s(h) + \frac{1}{2}d(\tilde{\mathcal{H}}\Delta\tilde{\mathcal{H}}) + \inf_{h \in \tilde{\mathcal{H}}} \varepsilon^t(h, \tilde{f}^s) + \varepsilon^s(h)$$

Finally, taking the infremum:

$$\inf_{h \in \mathcal{H}_\circ} \varepsilon^t(h) + \varepsilon^t(\tilde{h}, \tilde{f}^s \circ \varphi) \leq \frac{1}{2}\beta + \frac{1}{2}d(\tilde{\mathcal{H}}\Delta\tilde{\mathcal{H}}) + \inf_{h \in \mathcal{H}_\circ} \varepsilon^t(h) + \varepsilon^t(h) \tag{22}$$

Combining all inequalities leads to the stated one. $\qquad\square$

**Theorem 4** (Better control on the adaptability). $\tilde{\lambda}(\mathcal{H}_\circ) \leq \lambda(\tilde{\mathcal{H}})$.

*Proof.* We simply use the fact that $\tilde{\mathcal{H}} \subset \mathcal{H}_\circ$ then using the property of the infremum:

$$\tilde{\lambda}(\mathcal{H}_\circ) = \inf_{h \in \mathcal{H}_\circ} \varepsilon^t(h) + \varepsilon^s(h) \leq \inf_{h \in \tilde{\mathcal{H}}} \varepsilon^t(h) + \varepsilon^s(h) = \lambda(\tilde{\mathcal{H}}) \tag{23}$$

$\qquad\square$

**Theorem 5** ($\gamma$ embodies the risk of compression). *Let $(\varphi_1, \varphi_2) \in \Phi^2$ two source equivalent representations. If $\varphi_1$ is more compressed than $\varphi_2$, then $\gamma(\varphi_1, \mathcal{H}_\circ) \geq \gamma(\varphi_2, \mathcal{H}_\circ)$ for $\tilde{\mathcal{H}} \subset \mathcal{H}_\circ \subset \tilde{\mathcal{H}}^s$.*

*Proof.* If $\varphi_1$ and $\varphi_2$ are two equivalent representations, there is $\tilde{\mathcal{H}}(\varphi_1) \subset \tilde{\mathcal{H}}(\varphi_2) \subset \tilde{\mathcal{H}}^s$ then

$$\forall h \in \tilde{\mathcal{H}}^s, (\tilde{\mathcal{H}}(\varphi_1) \cap \{h' \in \mathcal{H} : \varepsilon^s(h, h') = 0\}) \subset (\tilde{\mathcal{H}}(\varphi_2) \cap \{h' \in \mathcal{H} : \varepsilon^s(h, h') = 0\})$$

which guarantees $\tilde{\mathcal{H}}_h(\varphi_1) \subset \tilde{\mathcal{H}}_h(\varphi_2)$. Finally

$$\varepsilon^t(h, \tilde{h}(\varphi_2)) \leq \varepsilon^t(h, \tilde{h}(\varphi_1))$$

invoking the property of the infremum where $\tilde{h}(\varphi) = \arg\min_{h' \in \tilde{\mathcal{H}}_h(\varphi)} \varepsilon^t(h, h')$. The supremum on $h \in \mathcal{H}_\circ$ conserves the inequality. $\qquad\square$

**Proposition 2** ($\beta$ is trainable from the data). *Let $(g_n, \varphi_n)_{n \in \mathbb{N}}$ a sequence such that $\alpha \cdot \tilde{\varepsilon}^s(g_n \circ \varphi_n) + (1 - \alpha) \cdot \mathbb{D}(\varphi_n) \to 0$ then $\beta_n \to 0$.*

*Proof.* First of all, since $\mathbb{D}(\varphi_n) \to 0$ then $\mathbb{P}^t(\varphi_n(X)) \to \mathbb{P}^s(\varphi_n(X))$. We note that for any $n \in \mathbb{N}$, $\beta \leq 2 \cdot (\varepsilon^s(g_n \circ \varphi_n) + \varepsilon^t(g_n \circ \varphi_n, \tilde{f}_n^s \circ \varphi_n)) \leq 2 \cdot (2 \cdot \varepsilon^s(g_n \circ \varphi_n) + (\varepsilon^t(g_n \circ \varphi_n, \tilde{f}_n^s \circ \varphi_n) - \varepsilon^s(g_n \circ \varphi_n)))$ Since $\varepsilon^s(g_n \circ \varphi_n) \to$ and $(\varepsilon^t(g_n \circ \varphi_n, \tilde{f}^s) - \varepsilon^s(g_n \circ \varphi_n)) \to 0$ (because $\mathbb{P}^t(\varphi_n(X)) \to \mathbb{P}^s(\varphi_n(X))$) there is $\beta_n \to 0$. $\qquad\square$

## B NOTATIONS

- $\Phi$ is the set of representations.
- $\mathcal{G}$ is the set of classifiers.
- $\mathcal{H}$ is the hypothesis space $\mathcal{H} = \mathcal{G} \circ \Phi$
- For a given $\varphi \in \Phi$, we note $\tilde{\mathcal{H}}(\varphi) = \{g \circ \varphi : g \in \mathcal{G}\} = \mathcal{G} \circ \varphi$. When there is no ambiguity, we note $\tilde{\mathcal{H}}(\varphi) = \tilde{\mathcal{H}}$.
- For a given $h \in \mathcal{H}$, we note $\varepsilon(h) = \mathbb{E}[(Y - h(X))^2]$. For $d = s, t$, we note a domain risk $\varepsilon^d(h) = \mathbb{E}^d[(Y - h(X))^2]$.
- For $d = s, t$ and a given $\varphi \in \Phi$, we note the conditional expectation $\tilde{f}^d(Z) = \mathbb{E}^d[Y|Z]$ with $Z = \varphi(X)$.

- **Equivalence.** For a given $\varphi \in \Phi$, the set of source equivalent representations $\tilde{\Phi}^s(\varphi) = \{\varphi' \in \Phi, \varphi' = \varphi \text{ on } \mathcal{D}^s\}$ (where $\mathcal{D}^s = \text{supp}(\mathbb{P}^s(X))$) and the source equivalent hypothesis space $\tilde{\mathcal{H}}^s(\varphi) = \{g \circ \varphi' : g \in \mathcal{G}, \varphi' \in \tilde{\Phi}^s\}$. Two elements of $\tilde{\Phi}^s$ are said source equivalent. When there is no ambiguity, we note $\tilde{\Phi}^s$ and $\tilde{\mathcal{H}}^s$.

- **Conservation.**
    - For a given $h \in \tilde{\mathcal{H}}^s$, we note $\tilde{\mathcal{H}}_h = \{h' \in \tilde{\mathcal{H}} : \varepsilon^s(h, h') = 0\}$.
    - For a given $h \in \tilde{\mathcal{H}}^s$, we note $\tilde{\mathbf{H}}_h = \{h' \in \tilde{\mathcal{H}} : h = h' \text{ on } \mathcal{D}^s\}$ where $\mathcal{D}^s = \text{supp}(\mathbb{P}^t(X))$.

- **Projection.**
    - $\pi : \tilde{\mathcal{H}}^s \to \tilde{\mathcal{H}}$ is defined as follows: $\pi(h) = \arg\min_{h' \in \tilde{\mathcal{H}}_h} \varepsilon^t(h, h')$ for $h \in \tilde{\mathcal{H}}^s$. $\pi(h)$ is called the projected hypothesis. If there is no uniqueness of the minimum, any solution works.
    - $\Pi : \tilde{\mathcal{H}}^s \to \tilde{\mathcal{H}}$ is defined as follows $\pi(h) = \arg\min_{h' \in \tilde{\mathbf{H}}_h} \varepsilon^t(h, h')$ for $h \in \tilde{\mathcal{H}}^s$. If there is no uniqueness of the minimum, any solution works.

