# OpenReview forum: "Domain-Invariant Representations: A Look on Compression and Weights"
_ICLR.cc/2020/Conference — Reject_

### Official Review · AnonReviewer2 · 2019-10-23
**Official Blind Review #2**

**Rating:** 3

**Review:**

This paper introduces the compression risk in domain-invariant representations. Learning domain-invariant representations leads to larger compression risks and potentially worse adaptability. To this end, the authors presents gamma(H) to measure the compression risk. Learning weighted representations to control source error, domain discrepancy, and compression simultaneously leads to a better tradeoff between invariance and compression, which is verified by experimental results.

The paper presents an in-depth analysis of compression and invariance, which provides some insight. However, I have several concerns:
* In Section 4, the authors propose a regularization to ensure h belongs to H_0. How is the regularization chosen? How does it perform on other datasets? Experimental results only on digit datasets are not convincing.
* In Section 5, the authors introduce weighted representations to alleviate the curse of invariance. However, they do not provide experiments to validate their improvement.
* The organization of this manuscript is poor and difficult to follow. Starting from Section 3, the authors use several definitions to introduce their main theorem. However, these definitions are somewhat misleading. I cannot get the point until the end of Section 3. Besides, the notations are confusing, so I have to go back to the previous sections in case of misunderstanding.





**Experience Assessment:**

I have published one or two papers in this area.

**Review Assessment: Checking Correctness Of Derivations And Theory:**

I assessed the sensibility of the derivations and theory.

**Review Assessment: Checking Correctness Of Experiments:**

I carefully checked the experiments.

**Review Assessment: Thoroughness In Paper Reading:**

I read the paper thoroughly.

---

> ### Author Response · Authors · 2019-11-13
> **Response to Reviewer2**
>
> Thank you for your comments and concerns.
>
> About the concerns on the experimental section and the choice of $\mathcal L_0(\varphi)$. This loss involves $\hat \varphi$, the representation for which we want to study both the risks of compression and adaptability. Then, we train $\varphi \in \Phi$ with the loss $-\Pi + \lambda_0 \cdot \mathcal L_0$ which is a penalized version of the constrained optimization $\min_{g\circ \varphi \in \mathcal H_0^\eta} -\Pi$. $\mathcal L_0$ involves two terms (equation 7). The first one enforces the source equivalence, the second one (with ReLU) enforces the new representations to not deviate from the original one in the target domain with a rate of $\eta$ (following the definition of $\mathcal H_0^\eta$).
>
> About the choice of the dataset and the scope of the experimental analysis. We followed a comparable experimental setup (both datasets and filtering strategies for studying the problem of target shift) than papers [1,2].
>
> [1] Zhao, Han, et al. "On Learning Invariant Representations for Domain Adaptation." International Conference on Machine Learning. 2019.
> [2] Johansson, Fredrik, David Sontag, and Rajesh Ranganath. "Support and Invertibility in Domain-Invariant Representations." The 22nd International Conference on Artificial Intelligence and Statistics. 2019.

---

### Official Review · AnonReviewer3 · 2019-10-23
**Official Blind Review #3**

**Rating:** 3

**Review:**

Summary
-------
This paper presents a revisit of existing theoretical frameworks in unsupervised domain adaptation in the context of learning invariant representation. They propose a novel bound that involves trainable terms taking into account some compression information and a novel interpretation of adaptability. The authors mention also contribution showing that weighting representations can be a way to improve the analysis.

Evaluation
-----
The ideas are novel and the result brings novel and interesting light on the difficult problem of unsupervised domain adaptation.
However, the practical interest in terms of applicability of the proposed framework is not fully demonstrated, the properties of the proposed analysis have to be studied more in details and some parts better justified. The experimental evaluation brings some interesting behavior but is somewhat limited. The weighting aspect of the contribution is not supported by any experiment.

Other comments
------------

-I am a but puzzled by the use of the term "compression". This is maybe subjective, but in the context of learning representation, I would have interpreted it as a way to sparsify the representation, and thus compression could then be measured with respect to a given norm (L2?) or another criterion (Kolmogoroff, ...).

In the paper, the notion of compression is related to a reduction of the hypothesis space after application of a transformation \phi, so I am wondering if using "hypothesis space reduction" would not be more appropriate.
In this case, however, there are maybe links with structural risk minimization that could be investigated here.
A side remark: there is no particular restriction on the space of transformations, we wonder if it would be useful to indicate if all the possible transformations are included as subspaces of a given latent space. Since, to be very general, one can imagine the existence of an unbounded number of transformations that correspond to an increase of the input dimension. For transformations leading to different representations of different dimensions, the way the deduced hypothesis can be compared should also be indicated (for defining properly the inclusion H(\phi_1)\subset H(\phi_2).

On the other hand, the authors seem to need the use of norms over transformations as illustrated in the definition of H_0^\eta in the experimental section. So I suggest that the analysis could be revisited by directly incorporating (representation) norms in the theoretical framework and in particular for defining more properly H_0.

-One weakness of the theoretical framework is for me the lack of definition of H_0 in Section 3. We just know that it is included between two classes of hypothesis of interest, but there is no clear characterisation of H_0 which makes the analysis fuzzy: we have a bound that involves an object without any clear definition and it is for me difficult to really interpret the bound. Trying to define H_0 with some restrictions related to the norm of the transformations, as evoked before, could be a way to address this point (and actually the way the experiments are done tend to confirm this point).

-Another weak point is the lack of qualitative analyse of the bound in Inequality 3 (the same applies for Inequality 5). I would have appreciated if the authors could provide an analysis similar to the one of (Mansour et al., COLT 2009) - it is cited in the paper - when they compared their result to the one of (Ben-David et al., 2007). For example, what happens when source is equal to the target, when is the bound significantly loose, significantly tight, different from other existing results, ...

In particular, if we compare the bound with the one of Ben-David et al. (we can also consider the one of Mansour et al.), there is two additional term, one is weighted by a factor 2, another one involved a supremum and one can think that this bound is rather loose and does not provide any insightful information and said differently it could not give a strong framework for practical considerations.
I may understand that when the bound is tight we could deduce that the compression term is low, but finding cases leading to a tight interesting bound does not seem obvious.

-The experimental evaluation presents some expected behavior in the context of the bound, but I miss a real study trying to make use of the proposed framework to do adaptation in practice with comparisons to other strategies.
Additionally, having additional studies with other models and tasks will probably reinforce the analysis.

-At the beginning of Section 3.2, the authors mention that they restrict their analysis to the square loss, however I think the analysis is true for larger class of losses with more general properties. In the experimental evaluation, the cross entropy is used, so I think that the experimental evaluation should also be consistent with the theoretical analysis by considering the square loss.


-Paragraph below Definition 5 is unclear: the notion of L2 norm has not been introduced in this context, so the message of the authors is a bit unclear.

-I do not find the notation \gamma(\phi,H) appropriate, I woud rather suggest to use \gamma(H\cdot \phi)

-The biblioggrgaphy can be improved by adding the right conferences/journals where the papers have been published in addition to the ArXiv reference.


**Experience Assessment:**

I have published in this field for several years.

**Review Assessment: Checking Correctness Of Derivations And Theory:**

I assessed the sensibility of the derivations and theory.

**Review Assessment: Checking Correctness Of Experiments:**

I assessed the sensibility of the experiments.

**Review Assessment: Thoroughness In Paper Reading:**

I read the paper at least twice and used my best judgement in assessing the paper.

---

> ### Author Response · Authors · 2019-11-13
> **Response to Reviewer3**
>
> We would like to thank you for your valuable comments.
>
> We provide details about your concerns. We follow the 'itemized' presentation for answering to each point.
>
> - (1 & 2nd items) We agree the term compression can be misleading and ‘hypothesis space reduction’ would be clearer. The relation of compression (which is defined as $\mathcal H(\varphi_1) \subset \mathcal H(\varphi_2)$ ie $\forall g \in \mathcal G, \exists g’ \in \mathcal G$ such that $g\circ\varphi_1 = g’ \circ\varphi_2$), we agree there is no reason to observe necessarily a relation of inclusion between two given representations. However, as mentioned by the reviewer, considering a set of possible transformations is relevant and deserves deeper investigations. For instance, one can expect to not reduce (so much) the hypothesis space if $\varphi_1$ is a transformation of $\varphi_2$ by a translation or a rotation. However, we believe this notion is in someway embedded in the definition itself: if $\mathcal G$ has enough capacity to ‘learn’ those transformations, then the raised point is addressed. Interestingly, this intuition is connected to the capacity of the set of classifiers $\mathcal G$.
>
> ---------------------------------------------
>
> - (3 & 4 th items) We underline that the bound is generally less tight than the original version [1]. In the particular case of $\mathcal H_0 = \tilde{\mathcal H}$, the bound is increased of $\frac 1 2 d(\tilde{\mathcal H}\Delta\tilde{\mathcal H}) + \beta$ with respect to [1]. When the size of $\mathcal H_0$ increases, our adaptability term decreases. But, this is followed by the increase in the term of compression (the trade-off introduced in the paper). To sum-up, we propose to better control the term of adaptability relying on a pessimistic estimation of the risk of compression (due to the supremum on $\mathcal H_0$).
>
> We would like to clarify the positioning of our work. DA bounds involve necessarily an intractable term during training (called adaptability in the literature relying on [1]). Roughly speaking, the tighter the bound, the more important becomes the contribution of intractable terms at training time. For instance, we can consider two limit cases (those cases are not relevant but show the difficulty) when considering the accuracy of a classifier:
> 1.  $\varepsilon^t(h) \leq \varepsilon^t(h)$: this bound is the tightest but is totally intractable.
> 2.  $\varepsilon^t(h) \leq 1$: tractable but not useful.
> Our analysis shows that we can reduce the importance of intractable terms in the bound by considering the risk of compression (this is a strategy and others should be also explored). The bound is significantly looser but more tractable and therefore offers better guarantees. The looseness of the bound can be controlled by the choice of a relevant $\mathcal H_0$ depending on the use case. In the submitted version, we provide the example $\mathcal H_0 = \mathcal H_0^\eta$ for addressing the robustness to adversarial attacks in a domain adaptation context.
>
> The main takeaways of our work (we are sorry for not having clearly stated it in the submitted version):
>
> “Achieving representation invariance may help to reduce the generalization gap between two domains. However, all invariances will not have the same guarantees for better generalization. We will look for representations which preserve the best the information in the original target features spaces. We leave both the design on new DA methods which incorporate such consideration and relevant design of $\mathcal H_0$ as future works.“
>
> -------------------------------------
>
> - In the experimental section, the cross-entropy is a trainable proxy for optimizing the accuracy. All error terms are reported on a test set computing the accuracy, not the cross-entropy, then the experimental analysis is consistent with the theoretical analysis. The choice of the squared loss is imposed by the use of the conditional expectation. More precisely, some parts of the proof need to bound $\varepsilon^s(h, \tilde f^s\circ\varphi) \leq \varepsilon^s(h)$ which derives from $\varepsilon^s(h) = \sigma^2 + \varepsilon^s(h, \tilde f^s\circ\varphi)$ where $\sigma^2$ is the noise in the data. For more general losses (which verify the triangular inequality), we have $\varepsilon^s(h) \leq  \sigma^2 + \varepsilon^s(h, \tilde f^s\circ\varphi)$ then no guarantee to have $\varepsilon^s(h, \tilde f^s\circ\varphi) \leq \varepsilon^s(h)$.
>
> ----------------------------------
>
> - We are sorry for the lack of clarity. Since we are studying a risk of a classifier $\hat Y = h(X)$ given by $\varepsilon(h) = \mathbb E [(Y - \hat Y)^2]$, we refer to this risk as a $L^2$ norm between estimated labels $\hat Y$  and true labels $Y$. Depending on the domain where such risk is computed, we refer to source or target $L^2$ norms.
>
> [1] Cortes, Corinna, Yishay Mansour, and Mehryar Mohri. "Learning bounds for importance weighting." Advances in neural information processing systems. 2010.

---

> > ### Comment · AnonReviewer3 · 2019-11-13
> > **Answer.**
> >
> > Ok, thank your for your answers.

---

### Official Review · AnonReviewer1 · 2019-10-24
**Official Blind Review #1**

**Rating:** 3

**Review:**

This submission provides a new theoretical framework for domain adaptation. In order to tackle the adaptability term in the classical domain adaptation theory, this submission proposes a new upper bound that enlarge the hypothesis space in the adaptability term. A weighted version of this theory is also given. Authors further support their conclusion by empirical results.

Pros:
1. This submission studies an important problem in domain adaptation.
2. This submission proposes new theoretical insight about compression and adaptability.
3. The conclusions of this paper can be partially proved by the empirical results.

Cons:
1.	As the author says in their future work, the source constraint is too strong that need to control the feature unchanged across all source domain. For this condition is not build on samples but on the support of source domain. It seems that authors use $L_0$ to constrain \phi’ to have same value with \phi on source dataset, which may be only a small part with zero measure of source support set.
2.	There is no generalization error analysis for these upper bounds. This submission provides weighted version of the main theory in the section 5. It seems that weighted version of upper bound could be further minimized by find a good weight. But add weight will add variance in the complexity term [A].
3.	This submission adds \beta term to change the adaptability term of $\tilde{\mathcal{H}}$ into the adaptability term of $\mathcal{H}_0$. The reason why \beta can be estimated from finite sample is not clarified, which is the premise of being trainable and should be mainly discussed in this paper. We can see that to estimate \beta is a domain adaptation problem under the fact that the labeled functions are same. \beta is a term that can’t be computed from small finite samples if there is no more assumption: It is not easy to approximate $\tilde{f}_s$ uniformly, otherwise the estimation of \beta will suffer from distribution shift. This submission claimed that the term can be trainable by giving Proposition 2, a proof of the consistency. However, this proposition is built on the assumption that there exists a series of \phi minimizing the distribution distance to zero. But this is impossible for finite sample estimation, when there will always be generalization error of estimating the distribution distance. Furthermore, it is usually impossible to make the two embedded distributions completely the same in empirical. In addition, if there is a series of \phi, how to control other terms in the upper bound? Every \phi will induce new $\tilde{\mathcal{H}}$ and $\mathcal{H}_0$ which will change all other terms. In summary, the main theory in this submission changes the unknown adaptability to a new term that is very hard to estimate. And there is no sufficient empirical or theoretical evidence in this paper that could support the fact that \beta is small. The contribution is therefore limited.
4.	The theory also fails to give upper bounds on compression term and adaptability term, or some explicit upper bounds for certain hypothesis spaces as examples. Readers could not have a clear image of how large will these terms be. Furthermore, if the support sets of source and target domain coincide a lot, the adaptability of $\mathcal{H}^s$ will not be too smaller than the previous one.
5.	The organization of the submission makes it hard to read:
a)	The symbol of this submission is chaotic. For example, $\tilde{\mathcal{H}}$, $\tilde{\mathcal{H}}^s$ ,$\tilde{\mathcal{H}}_h$ are defined based on $\phi$, $\pi(h)$ is defined based on $\tilde{\mathcal{H}}$, but all these facts are not revealed in their symbols.
b)	For clarity, all loss functions defined in Section 4 should be stated in a independent line. The Section 5 should be moved to the front of Experiment part.
c)	I recommend the authors to restate all new defined symbols as a list on the top of appendix. It really troubles me during checking the proof.

I think this submission discusses about an important problem and provides new insight, but it is not a thorough theoretical work because of above reasons. So, I vote for rejecting this paper.

[A] Cortes, Corinna, Yishay Mansour, and Mehryar Mohri. "Learning bounds for importance weighting." Advances in neural information processing systems. 2010.


**Experience Assessment:**

I have published one or two papers in this area.

**Review Assessment: Checking Correctness Of Derivations And Theory:**

I carefully checked the derivations and theory.

**Review Assessment: Checking Correctness Of Experiments:**

I assessed the sensibility of the experiments.

**Review Assessment: Thoroughness In Paper Reading:**

I read the paper thoroughly.

---

> ### Author Response · Authors · 2019-11-13
> **Response to Reviewer1**
>
> We thank you for the fruitful comments and suggestions.
>
> Our responses are below:
>
> 1.  The source equivalence is a strong constraint. Understanding how this constraint can be implemented in a finite sampling case will provide a broader range of application of our analysis. In the experimental section, we suggest to implement the source equivalence with the loss $\mathbb E^s[||\varphi'(X) - \varphi(X)||^2]$.
>
> 2. Thanks for pointing this issue when extending the analysis to finite sampling. We agree with the fact that adding weights will increase the variance (proportionally with $\left (\mathbb E^s [w^2] \right)^{1/2}$) of the complexity term when bounding empirical errors. However, our weighting strategy is flexible enough for controlling the variance of $w$ (for instance by enforcing $w\leq M$ for a given $M>0$ in the choice of $\mathcal W$).  In the section 5 of [1], they suggest to rely on alternative weight $u$ which achieves a trade-off between re-weighting ($u\approx \mathbb P^t(X)/\mathbb P^s(X)$) losses and and high variance (preventing too high value of  $\left (\mathbb E^s [u^2] \right)^{1/2}$). In our work, the trade-off is between relaxing the constraint of invariance and limiting the weight variance. This trade-off is embedded in the choice of $\mathcal W$. We will provide in the updated version a clear detail of the interest of adding weight for invariance relaxation.
>
> 3. Thanks for pointing out this interesting issue. We agree that $\beta$ is a domain adaptation problem by itself but in a covariate shift situation in the representation space (indeed, label function is the same for both domains: it is $\tilde f^s$). Therefore, one can expect this term to be small since the infremum on $\tilde{\mathcal H}$ is involved. Another way to deal with it is to input $\hat f^s = \arg \min_{h \in \tilde{\mathcal H}} \varepsilon^s(h)$ in the inequality rather than $\tilde f^s$ (this leads to some but doable changes in the inequality).
>
> About the sequence of $(\varphi_n)$ introduced for demonstrating that $\beta$ is trainable from the data. First of all, if $\varphi$ changes, all terms in the bound change as mentioned by the reviewer. More importantly, the adaptability term $\lambda(\tilde{\mathcal H}) = \inf_{h \in \tilde{\mathcal H}} \varepsilon^s(h) + \varepsilon^t(h)$ from [2] change. This is the setting of methods based on Domain Adversarial Learning. We have shown that learning invariant representations controls $\beta$. To do so, we have considered a sequence of representations which  converges to 0 in the sense of Learning Objective 1 (the objective of Domain Adversarial Learning). If such condition hold, then $\beta$ tends to 0 also. In other words, we have shown that: $\varepsilon^s(h) + d(\tilde{\mathcal H}\Delta\tilde{\mathcal H}) + \beta  = \varepsilon^s(h) + d(\tilde{\mathcal H}\Delta\tilde{\mathcal H}) + o(\varepsilon^s(h) + d(\tilde{\mathcal H}\Delta\tilde{\mathcal H}))$. The distribution may mismatch but $\beta$ is neglectable with respect to the term which quantifies distribution mismatching. This is the reason why we considered it as a minor issue during our analysis.
>
> 4. If the source and the target supports coincide a lot (let consider the  limit case they are equals), the source conservation enforces $\tilde{\mathcal H}^s = \tilde{\mathcal H}$ (which implies a compression term equals to 0) and the bound is increased by $\frac 1 2 d(\tilde{\mathcal H}\Delta\tilde{\mathcal H}) + \beta$ with respect to the one introduced in [2].
>
> 5.a In the notation, we emphasize the dependence on $\varphi$ with $\tilde{\cdot}$. Thank you for raising the difficulty of reading the notations, we will think about an other notation system.
>
> 5.b We will provide an updated version of the manuscript with a clear statement of the losses.
>
> 5.c We will provide a notation page in the appendix
>
>
>
>
> [1] Cortes, Corinna, Yishay Mansour, and Mehryar Mohri. "Learning bounds for importance weighting." Advances in neural information processing systems. 2010.
> [2] Ben-David, Shai, et al. "A theory of learning from different domains." Machine learning 79.1-2 (2010): 151-175.

---

### Author Response · Authors · 2019-11-15
**Updated version of the submission**

We would like to thank the reviewers for their valuable and insightful comments which have helped us to improve our submission and for their time for checking both the proofs and experiments.

We provide an updated version  of the submission which addresses some concerns of the reviewers:
- We clarify the experimental section bringing more details on the choice of the different losses (ask by reviewer1).
- We add a comment on the risk of increasing the variance of estimators when using weights in section 5 (a concern of reviewer1).
- We extend our discussion part by indicating precisely the scope of this work.
- We add a notation section in the appendix which includes the most important ratings and definitions (ask by reviewer1)
- We correct the bibliography by adding the right conferences/journals where the papers have been published in addition to the ArXiv reference (ask by reviewer3).

---

### Decision · Program_Chairs · 2019-12-19

**Decision:**

Reject

**Comment:**

This paper provides a new theoretical framework for domain adaptation by exploring the compression and adaptability.

Reviewers and AC generally agree that this paper discusses about an important problem and provides new insight, but it is not a thorough theoretical work. The reviewers identified several key limitations of the theory such as unrealistic condition and approximation. Some important points still require more work to make the framework practical for algorithm design and computation. The presentation could also be improved.

Hence I recommend rejection.